# Perceived Stress in Relation to Quality of Life and Resilience in Patients with Advanced Chronic Kidney Disease Undergoing Hemodialysis

**DOI:** 10.3390/ijerph18020536

**Published:** 2021-01-11

**Authors:** Pedro García-Martínez, Rafael Ballester-Arnal, Kavita Gandhi-Morar, Jesús Castro-Calvo, Vicente Gea-Caballero, Raúl Juárez-Vela, Carlos Saus-Ortega, Raimunda Montejano-Lozoya, Eva María Sosa-Palanca, María del Rosario Gómez-Romero, Eladio Collado-Boira

**Affiliations:** 1Nursing School La Fe, Adscript Center of Universidad de Valencia, 46026 Valencia, Spain; garcia_pedmarb@gva.es (P.G.-M.); saus_car@gva.es (C.S.-O.); montejano_rai@gva.es (R.M.-L.); sosa_eva@gva.es (E.M.S.-P.); gomez_ros@gva.es (M.d.R.G.-R.); 2Research Group GREIACC, Health Research Institute La Fe, Avda. Fernando Abril Martorell, 106. Pabellón docente Torre H, Hospital La Fe, 46016 Valencia, Spain; 3Faculty of Health Sciences, University Jaume I, 12071 Castellon, Spain; rballest@psb.uji.es (R.B.-A.); kavitagandhimorar@gmail.com (K.G.-M.); colladoe@uji.es (E.C.-B.); 4Department of Personality, Assessment, and Psychological Treatments, University of Valencia, 46010 Valencia, Spain; 5Department of Nursing, University of La Rioja, 26006 Logroño, La Rioja, Spain; raul.juarez@unirioja.es

**Keywords:** chronic kidney disease, perceived stress, resilience, quality of life

## Abstract

*Background*: Patients with chronic kidney disease undergo various stages of therapeutic adaptation which involve lifestyle modifications, physical changes, and adjustment to renal replacement therapy. This process produces adaptive stress. *Objective:* To identify how resilience, health- related quality of life, and sociodemographic, clinical, and hemodialysis routine-related variables are related to perceived stress in patients with chronic kidney disease receiving hemodialysis for more than six months. *Methods:* This was a multicenter and cross-sectional study involving 144 patients from the Valencian Community (Spain). The assessment scales used for the study were the Perceived Stress Scale 10, the Kidney Disease Quality of Life 36, and the Connors–Davidson Resilience Scale. To identify variables with predictive power over Perceived Stress Scale 10 scores, multiple regression analyses were performed. *Results:* Employment status (*p* = 0.003), resilience (*p* < 0.001), and quality of life (*p* < 0.001) were shown to be significantly related to perceived stress. The regression models determined that health-related quality of life and resilience explained up to 27.1% of the variance of total PSS10 scores. *Conclusions:* Resilience was identified as one of the most important predictors of Perceived Stress Scale 10 scores. Thus, the development of interventions to promote resilience may have a positive impact on perceived stress in patients with chronic kidney disease.

## 1. Introduction

Chronic kidney disease (CKD) is a public health issue that affects over 10% of the global population [1,2]. The worldwide prevalence of CKD is estimated to vary between 7% in Asia and 12% in Europe. In Spain, the prevalence of CKD is 13%. Of these patients, 41% are treated using renal replacement therapy (RRT) with hemodialysis (HD) [3].

Therapeutic management of the HD patient is based on pharmacological measures as well as lifestyle modifications or changes in routine that include healthy habits, visits to the health center for HD, and vascular access care [4]. HD patients may experience unwanted physiological symptoms such as dry skin, itching, tiredness, or loss of energy, which can affect activities of daily living and self-care [5,6]. Negative psychological and social effects such as role alterations due to reduced emotional well-being, changes in body image, and increased time dedicated to self-care and RRT also commonly occur [7]. All these symptoms can affect the perception of stress associated with the disease and perceived health-related quality of life (HRQoL).

Perceived stress occurs when the individual considers their relationship with the environment to be threatening or overwhelming in a way which can affect well-being [8]. The HD patient experiences periods of great psychological stress over the course of the disease, with adaptation becoming necessary [9,10]. This adaptation process usually takes place over around 66 days [11]. After six months of HD and with new consolidated care habits, the patient should be able to face the illness with low levels of stress, as the perception of stress is dependent on life events and the development of coping strategies that form part of the concept of resilience.

Resilience is defined as a coping style that allows for positive progress in adverse situations [12]. The resilient coping style involves innate factors, resilience-related traits (such as self-esteem, self-efficacy, and support networks), and finally, a personal reintegration process [12]. In CKD patients with more favorable outcomes, resilience has been found to be an associated protective factor [13]. 

Among the models aimed at understanding the fundamental elements of resilience is the Systematic Self-Reflection (SSR) model. This model differentiates three basics concepts for coping: (a) resilient resources, (b) their use and emotional regulation, and (c) resilient beliefs [14]. Resilient resources in HD include the cognitive, employment-related, family, and economic resources used to cope with the disease.

Finally, in HD patients, quality of life can be affected by symptoms related to the end stage of CKD. HRQoL refers to how well a person functions in life and his or her perceived well-being in the physical, mental, and social domains of health [15]. HRQoL has been identified as a predictor of the effects and course of chronic disease and can be used as a measure to assess treatment and facilitate patient decision-making [16].

One way of delaying the evolution of the disease and reducing the risk of these complications is the use of tools to reach a consensus in decision-making between the medical team and the patient. This interaction between patients and health professionals improves risk perception, reduces the levels of conflict in decision-making, and increases the sense of control perceived by the patient [4]. The educational process for information used as a consensus tool is started in nephrology consultation [4]. However, in some countries, such as Uruguay [8] and Spain the educational process can be initiated by primary care nurses and continues with a specialized consultation in which nurses play a major role [4]. Differences between countries in the educational process of the patients may be due to the resources and organization of each health system [4], but in all cases, its objectives include to promote coping skills and strategies that are associated with a better evolution of CKD, increasing adherence to treatment and health-related quality of life (HRQoL) [9,10]. 

In this sense, HRQOL refers to how well a person functions in their life and his or her perceived well-being in physical, mental, and social domains of health [11]. HRQoL has been identified as a predictor of the effects and evolution of chronic disease, allows for the evaluation of treatment, and facilitates patient decision-making [12].

The HD patient experiences three periods of great psychological stress during the progression of the disease: (1) at the time of diagnosis of CKD, (2) at the time of diagnosis of ACKD and (3) when to start the RRT, and consequent adaptation of lifestyle [13,14]. Adaptation process, if appropriate, usually takes about sixty-six days [15]. So, after 6 months of starting HD, with the new consolidated care habits, the patient should face the illness with a low level of stress as the perception of stress is shown to be a variable dependent on life events and the development of coping strategies. The ability to develop these coping resources is a key element in building the resilience of these patients; resilience is defined as a coping style that allows positive progress in adverse situations [16] and is associated with a better evolution of CKD [9]. The resilient coping style involves innate factors, resilience-related traits such as self-esteem, self-efficacy, and support networks, and finally a personal reintegration process [16].

Controversy currently exists surrounding the relationship between psychological stress and HRQoL in adaptation to chronic illness. Some authors have shown how adaptation to the disease and its treatment reduces stress levels and increases HRQoL [17,18], while other authors have not found this relationship [19]. In our setting, it has been proposed that resilience is key to adjusting to stressful situations and, therefore, to chronic illness [20,21]. This controversy justifies an in-depth study of the factors that influence the life process of HD patients to improve our understanding of the relationships between psychological stress, resilience, and HRQoL. Clarifying the phenomenon will allow for the development of individual strategies that can contribute to improving quality of life and reducing health expenditure in an aging population. The average HD patient is 64.5 years of age and has a survival time of 6.3 years [3]; the mean annual cost of HD is 54,527.29 USD per patient [22].

For the aforesaid reasons, this study aimed to identify how resilience, health-related quality of life, and sociodemographic, clinical, and hemodialysis routine-related variables are related to perceived stress in patients with chronic kidney disease undergoing hemodialysis for more than six months. A secondary objective was to identify how the studied variables could predict perceived stress in these patients. 

## 2. Materials and Methods 

### 2.1. Design

Multicenter and cross-sectional study carried out in the Valencian Community (Spain) in 2016.

### 2.2. Participants 

The sample population was selected from all ACKD patients in the Valencian Community receiving hospital-based hemodialysis from the renal care service provider Diaverum. Five of the eight hemodialysis clinics in the area managed by Diaverum agreed to participate in the study. These five clinics provided treatment to a total of 407 patients. All patients meeting the inclusion criteria were asked to participate. 

Of the 407 patients treated at the Diaverum clinics, 60 patients were treated in three small clinics that chose not to participate in the study, 54 had received treatment for less than six months, 42 presented with cognitive problems, and 28 had problems with reading and writing in Spanish. As a result, 223 potential participants for the study remained. To achieve a representative sample of the five clinics with a 95% confidence level, a 5% of error rate, and assuming the greatest possible heterogeneity (50%), a minimum sample size of 142 patients was calculated. Data collection took place in January 2016.

To be included in the study, participants were required to be CKD patients aged over 18 who had received HD for over six months. Patients with literacy or cognitive problems that limited self-completion of the questionnaires were excluded.

### 2.3. Variables and Instruments 

Sociodemographic, HD Routine-Related, and Clinical Data: An ad hoc questionnaire was created to collect data. The sociodemographic variables included age, sex, level of education, employment situation, home setting, and living arrangement, whereas the HD routine-related variables included means of transport and accessibility to the clinic. Data on clinical variables were recovered from patient files and included information on diabetic status, type of vascular access used for treatment (arteriovenous fistula/graft or central catheter), and comorbidity. 

Perceived Stress: The Perceived Stress Scale 10 (PSS10) is composed of 10 items, with scores ranging between 0 (lower stress level) and 40 points (maximum stress level). The Spanish version of the PSS10 has a Cronbach’s alpha of 0.82 [23].

Resilience: The Connor–Davidson Resilience Scale (CD-RISC) questionnaire features 25 items and has a score range between 0 (lower resilience level) and 100 (maximum resilience level). The Spanish version of the CD-RISC has a Cronbach’s alpha of 0.89 [24]. 

Health-Related Quality of Life: The Spanish version of the Kidney Disease Quality of Life 36 (KDQOL-36) was used (36 items) [25]. This assessment tool includes the generic SF-12 questionnaire and three specific subscales for ACKD patients relating to: (a) burden of kidney disease; (b) symptoms/problems of kidney disease (symptoms/problems); and (c) effects of CKD on daily life (effects of CKD). Higher scores on these scales reflect a better HRQoL [25]. The Spanish version of the scales has a Cronbach’s alpha of >0.90 [26]. 

### 2.4. Data Analysis

To present descriptive information on the sample, the mean (M), the standard deviation (SD), the median, and the interquartile range of the quantitative variables were used, as well as the frequencies and percentages of the qualitative variables. 

The internal consistency for the PSS10, the CD-RISC, and the KDQoL-36 was assessed using the Cronbach’s alpha coefficient. 

The normality analysis of the quantitative variables was performed using the Kolmogorov–Smirnov and Shapiro–Wilk normality tests. For the bivariate analysis with dichotomous variables the independent samples *t*-test was used, and ANOVA was employed for polytomous variables. The Bonferroni post hoc test was implemented in the case of significant differences in the ANOVA. Given the non-normal distribution of all variables except age, correlations were determined using Spearman’s rho.

Finally, multiple regression analysis was performed using the forward stepwise method. Only normally distributed variables were used as dependent variables. Among the models obtained, the parsimony principle was applied [27]. Given our limited sample size and the non-normal distribution of independent variables, residual errors from the resulting models were inspected to ensure their normal distribution and thus the reliability of our regression models [28]. To identify the predictive value of the model, the Cohen criterion [29] was applied to one-way ANOVA models. 

Statistical analysis was performed using SPSS v23 (Armonk, NY, USA) (statistical significance was considered at *p* = 0.05).

### 2.5. Ethical Considerations 

The study was approved by the Ethical Committees at Jaume I University of Castellón (Spain) and was conducted in accordance with Spanish regulations for the protection of personal data.

## 3. Results

In total, 223 patients were invited to participate in the study. Of these, 64.6% completed all the assessment scales, resulting in a final study sample of only 144 patients.

The mean age of patients was 67.31 years old (*SD* = 12.32). Of the participants, 68.8% were male, 29% were diabetic, and 83.2% had undergone arteriovenous vascular access placement. A significant difference was observed between the groups with regard to employment status and perceived level of stress (*p* = 0.003) (Table 1). Bonferroni’s test showed that perceived stress was significantly higher among the unemployed and homemakers as compared to the actively employed (*p* = 0.025) or the incapacitated/retired (*p* = 0.012). No significant difference in perceived stress was found between actively employed participants and the incapacitated or retired (*p* = 0.276).

Stress (as per the PSS10) correlated negatively with resilience (ρ = –0.404) and HRQoL (ρ = –0.368). Age (ρ = 0.022) and comorbidity (ρ = −0.003) were not significantly correlated with stress (Table 2). 

We developed different linear regression models to determine which variables had predictive power over PSS10 scores (Table 3). In these models, the variables that had shown a significant relationship with the PSS10 were retained. The variables thus introduced as predictive variables were: (a) Employment status, (b) resilience (CD-RISC), and (c) HRQoL (KDQoL-36). 

The regression model for the PSS10 (Table 3) showed how the variables of HRQoL and resilience explained up to 27.1% (R^2^ = 0.271) of the variance of PSS10. After testing the normality of the residues in the regression analysis, a normalized distribution was observed (*p* = 0.200).

## 4. Discussion

The main objective of this study was to identify variables affecting perceived stress in HD patients with ACKD. It was found that HRQoL and resilience constituted the main predictors of perceived stress at this stage of the disease. 

The finding that HRQoL and resilience are the main predictors of perceived stress is in line with previous studies, which argue that resilience is key to adjusting to stressful situations in chronic disease [20,21]. Thus, resilience acts as a protective factor, as it improves the capacity of the individual to cope with disease psychologically [30], acts as a mediator between personal history and adaptation to chronic disease [31] and is the main predictor of HRQoL in HD patients [32]. 

Alterations in risk and protective factors occur during stressful circumstances related to the disease [21]. This process of adaptation results in changes in the impact of these factors on psychological variables such as resilience or HRQoL [20]. Thus, positive coping during the initial stages of CKD will result in greater resilience, which can improve HRQoL [33] and reduce perceived stress in later stages of the disease.

The SSR model helps to explain the relationship between perceived stress, health-related quality of life, and resilience. Resilient resources, emotional regulation strategies (which modulate perceived stress), and resilient beliefs (perceived quality of life) aid in the management of stressful situations. Once the stressful event has been overcome, in the short term the use of resources and new emotional regulation strategies can modify perceived quality of life. In the medium term and after a period of self-reflection, resilient resources, their use, and resilient values are redefined, ultimately modifying resilience [14]. Finally, stable resilient resources (sex, age, living arrangement, educational level, or home setting) will then have a reduced influence on stress with each new situation. This could explain the absence of significant relationships found between stress and other socio-demographic, HD routine-related, or clinical variables. 

Of the socioeconomic, clinical, and HD routine-related factors studied in our sample of HD patients with ACKD, only patient employment status was found to be significantly related to perceived stress. Seventy-eight percent of patients depended on financial resources such as retirement or unemployment benefits. This figure is higher than that reported for patients undergoing other types of RRT such as ambulatory peritoneal dialysis, where 60% of individuals were found to be professionally inactive [34]. In this study, those who remained active in the workplace (with very low representation in our sample) and above all retired individuals had lower levels of perceived stress than the unemployed or those engaged in domestic work, possibly due to better income (typically a protective factor against stress) and higher levels of activity and socialization (also protective against stress) [34]. The loss of resources has been found to be related to higher levels of perceived stress [35,36,37,38], emphasizing the importance of an intersectoral approach and social interventions to reinforce possible unfavorable situations in these patients [39]. With the results obtained in our sample, it was not possible to explain the fact that while employment status influenced stress, it did not affect resilience and quality of life. 

No significant differences were found with regard to adaptation to changes in daily routines, the need for personal support, and transport to the clinic, in line with the proposed adaptation period of 66 days [11].

In addition, no significant differences were found for the analyzed clinical variables in relation to perceived stress, unlike other studies where it has been suggested that people with greater comorbidity or poorer clinical conditions tend to present with higher stress levels [35,40,41]. Nonetheless, our data are consistent with the dynamic resilience model [20,21], according to which if the result of coping is positive, greater adaptation to chronic disease will occur, reducing stress and increasing HRQoL [17,18]. 

In summary, the absence of sociodemographic, clinical, and HD routine-related factors associated with perceived stress within the framework of the dynamic model of resilience [20,21] could be related to the multistage nature of CKD [4], and to adaptation through traits of resilience and reintegration processes in the coping style proposed by Richardson [12] within the transactional theory of stress [8]. These findings again support that resilience and HRQoL are the main predictors of perceived stress. 

This study provides support for patient involvement strategies in educative processes for therapeutic decision-making, as well as multidisciplinary and group-based education for CKD patients [42,43] to increase self-esteem and self-care and provide resources for resilience [12]. In this context, consensus decision-making as well as adaptation and lifestyle modifications are key to stress management and better disease control [4]. To carry out these interventions, several action models have been proposed. In this context, the expert patient program developed in Spanish hospitals should be highlighted. In this program, peer education is carried out by sick individuals who possess high levels of knowledge with regard to their illness, good leadership and communication skills, and strong tools for psychological management of changes associated with the disease. This expert patient program has already shown positive results with regard to patient satisfaction, knowledge, skills, and quality of life [44]. Mindfulness-based stress reduction (MBSR) programs have also been shown to provide benefits and improve self-efficacy associated with aspects of behavior and lifestyle in chronic disease [45,46,47,48]. Following the advanced practice nursing model [49], the application of MBSR is proposed for use by highly specialized CKD nurses as part of practice to impart theoretical and practical knowledge on stress, meditation, breathing techniques, yoga, mindfulness, or relaxation adapted to patients at each stage of the disease [50]. 

There is a clear need for the development of studies with new designs to clarify the problem and provide stronger evidence on how resilience and adaptation are associated with stress through cohort studies. If this relationship is confirmed, the effectiveness of interventions should be assessed through clinical trials, either with the expert patient model or the advanced practice nursing model in specialized CKD practice.

**Limitations:** Several limitations must be acknowledged in this study. Our study included a larger number of participants as compared to other studies involving ACKD patients (e.g., *n* = 165 [51], *n* = 103 [52], or *n* = 37 [53]). However, even considering the difficulties in recruiting patients matching our strict inclusion criteria, we are aware that the sample size is limited. Thus, we cannot ensure that our results represent the global population of patients with ACKD, limiting the generalization of the results. In addition, as this is a transversal study, we cannot assure a cause-and-effect relationship between the variables. However, the results are consistent with the SSR model [14]. 

## 5. Conclusions

Resilience and HRQoL were found to be the main predictors of perceived stress among patients undergoing HD for more than six months. Among the sociodemographic factors analyzed in this study, only employment status was found to be related, with greater perceived stress being observed in the unemployed and in those engaged in domestic work. The remaining sociodemographic, HD routine-related, and clinical variables were not shown to be related to perceived stress in this population.

## Figures and Tables

**Table 1 ijerph-18-00536-t001:** Sample distribution (%) and average PSS10, CD RISC, and KDQoL36 (M) scores in relation to sociodemographic, hemodialysis (HD) routine-related, and clinical data.

Variables	Sample Distribution	PSS10	CD RISC	KDQOL-36
Sex *(t*-test *p*-value)		0.165	0.291	0.707
Male	68.8%	12.36	71.42	47.43
Female	29.2%	14.33	67.15	46.40
Living arrangement (ANOVA *p-*value)		0.150	0.281	0.858
With a partner	42.4%	11.89	73.04	47.22
With a partner and children	22.9%	14.21	68.57	48.33
With children	6.9%	17.90	68.08	44.68
With other family members	9.7%	14.07	63.02	51.00
With caregivers	4.2%	15.50	56.41	45.67
Education level (ANOVA *p-*value)		0.316	0.291	0.628
No studies	9.0%	12.23	65.09	47.71
Primary school	34.0%	14.82	66.63	45.51
Secondary school	28.5%	11.98	75.69	49.63
Vocational training	15.3%	13.45	68.22	48.01
University studies	10.5%	10.93	72.17	44.38
Employment status (ANOVA *p-*value)		0.003 *	0.128	0.053
Actively employed	1.4%	3.5	88.46	67.98
Unemployed/homemaker(Un/ho)	10.4%	18.47	61.28	47.72
Incapacitated or retired	84.1%	12.50	70.51	46.53
Home setting (*t*-test *p*-value)		0.598	0.236	0.683
Rural	14.6%	13.95	64.13	46.49
Urban	71.5%	12.97	70.45	48.06
Type of transport (ANOVA *p-*value)		0.455	0.151	0.128
Private	31.3%	12.00	73.79	50.91
Public	16.0%	13.35	63.38	44.29
Medical transport	41.0%	13.92	68.08	46.42
Perception of displacement (*t*-test *p*-value)		0.129	0.257	0.798
Easy or very easy	59.8%	12.01	71.84	48.36
Normal	34.6%	14.66	65.70	46.63
Difficult or very difficult	4.7%	15.71	63.19	46.53
Diabetic status *(t*-test *p*-value)		0.619	0.646	0.183
Diabetic	29.2%	13.24	69.57	44.73
Non-diabetic	66.7%	12.53	71.40	48.,29
Type of vascular access (*t*-test *p*-value)		0.323	0.015 *	0.368
Arteriovenous fistula/graft	84.0%	12.60	72.19	47.74
Tunneled central venous catheter	13.9%	14.47	59.16	44.58

Note: PSS10 = Perceived Stress Scale 10; CD-RISC = Connor–Davidson Resilience Scale; KDQoL36 = Kidney Disease Quality of Life 36 total score. * *p* < 0.05.

**Table 2 ijerph-18-00536-t002:** Correlations between the study measures, means, standard deviations, interquartile ranges, and normality tests.

Variable	PSS10	CD-RISC	KDQoL-36	Age	Comorbidity
CD-RISC	−0.404 *				
KDQoL-36	−0.368 *	0.381 *			
Age	0.022	−0.072	−0.385 *		
Comorbidity	−0.003	−0.059	−0.278 *	0.622 *	
Mean	12.93	70.08	47.19	67.31	6.04
SD	7.669	21.037	12.536	12.327	2.079
Interquartile range	11.00	31,731	18,979	18.00	4.00
Kolmogorov–Smirnov *p*-value	0.180	0.001	0.034	0.145	0.001
Shapiro–Wilk *p*-value	0.005	0.000	0.066	0.204	0.012

Note: PSS10 = Perceived Stress Scale 10; CD-RISC = Connor–Davidson Resilience Scale; KDQoL36 = Kidney Disease Quality of Life 36 total score; Comorbidity = Charlson comorbidity index. * *p* < 0.001.

**Table 3 ijerph-18-00536-t003:** Linear regression models for the PSS10.

	β	*p*	R^2^	F
Dependent variable: PSS10				
Model information		<0.001	27.1%	20.732
KDQoL36	−0.205	<0.001	18.9%	
CD-RISC	−0.116	0.001	8.7%	

Note: PSS10 = Perceived Stress Scale 10; CD-RISC = Connor–Davidson Resilience Scale, KDQoL36 = Kidney Disease Quality of Life 36 total score.

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
