# Peer review of "Perceived Stress in Relation to Quality of Life and Resilience in Patients with Advanced Chronic Kidney Disease Undergoing Hemodialysis"

_ijerph, 2021, doi:10.3390/ijerph18020536_

Round 1
Reviewer 1 Report
García Martínez P et al. evaluated the relationship between perceived stress, resilience, and health-related quality of life as well as with others sociodemographic, clinical, and daily therapeutic routine variables of hemodialysis. The topic is interesting; however, the results the results are not enough to support the conclusion.
Major:
- Introduction is too long. Second paragraph looks not necessary.
- I note that all arteriovenous shunt are fistulas. Are the authors sure that no graft in the studies patients?
- I think the big problem in the article is that the statistical analysis could not support the conclusion. No multivariable adjustment. No laboratory data of these HD patients in the analysis (lab data would influence the stress, resilience….)
- Minor:
- Please use ”HD” directly, no need to use CKD patients undergoing HD or CKD therapy with HD.
- No ACKD full name in the introduction.
- Hemodialysis (full name) or chronic kidney disease (full name) are shown in 2.2
- The presentation of the table 1 is difficult to understand. In addition, please show post-hoc-analysis in the table.
Author Response
REV1
Major:
Dear Reviewer,
Thank you for taking the time to review this manuscript. We greatly appreciate your feedback and are aware that with your comments and suggestions the manuscript has been significantly improved. We have addressed your concerns and hope you will consider this manuscript for publication.
This manuscript has been reviewed by a professional translator in order to improve and facilitate the understanding of the article. We hope that the resulting changes are to the liking of the reviewers.
Introduction is too long. Second paragraph looks not necessary.
We have revised the introduction to reduce its length and have deleted the second paragraph, as indicated in your assessment.
I note that all arteriovenous shunt are fistulas. Are the authors sure that no graft in the studies patients?
In the collection of clinical data, a distinction was made between arteriovenous fistulas/grafts and central catheters. However, no such distinction was made between arteriovenous fistulas and grafts in data collection. As a result, we do not have access to this data.
To avoid confusion we have introduced the following clarification:
¨Data on clinical variables were recovered from patient files, and included information on diabetes mellitus, type of vascular access used for treatment (arteriovenous fistula/graft or central catheter), and comorbidity.¨
I think the big problem in the article is that the statistical analysis could not support the conclusion. No multivariable adjustment. No laboratory data of these HD patients in the analysis (lab data would influence the stress, resilience….)
With regard to the lack of support from statistical analysis for the conclusions, we have provided the following explanation in the limitations:
¨In addition, as this is a transversal study, we cannot assure a cause-and-effect relationship between the variables. However, the results are consistent with the SSR model [14]¨
Furthermore, this study did not intend to assess patient stress based on analytical parameters, so they were not used for this study. We have attempted to determine how some variables (particularly resilience and quality of life) affect perceived stress, and the study design and analysis were proposed to this end.
Minor:
Please use ”HD” directly, no need to use CKD patients undergoing HD or CKD therapy with HD.
Thank you for your suggestion. This has been modified in the original text
No ACKD full name in the introduction.
We have introduced the following clarification: The patient with advanced chronic kidney disease (ACKD) undergoing HD can cause may experience unwanted physiological effects symptoms such as dry skin, itching, tiredness, or loss of energy, which can affect activities of daily living and self-care [5,6]
Hemodialysis (full name) or chronic kidney disease (full name) are shown in 2.2
We accept your suggestion and have modified it in the original text
The presentation of the table 1 is difficult to understand. In addition, please show post-hoc-analysis in the table.
Changes have been made to the table to make it easier to read. The post hoc tests have been explained as follows:
¨A significant difference was observed between the groups with regard to employment situation and perceived level of stress (p=0.003). Bonferroni´s test showed that the perceived stress was significantly higher among those who were unemployed or homemakers as compared to the actively employed (p=0.025) or those who were disabled/retired (p=0.012). No significant difference in perceived stress was found between the actively employed and the disabled or retired (p=0.276).¨
Reviewer 2 Report
Garcia-Marinez and authors carried out a multicenter, cross-sectional and correlational study to identify the relationship between perceived stress, resilience, and health-related quality of life in patients with chronic kidney disease, undergoing Hemodialysis for more than six months.
It is an very interesting study with the potential to provide some very important information which could lead to positive outcomes for chronic kidney diseases patients. However the report only identified significant correlations between employment status, resilience, quality of life and perceived stress while many other variables showed no impact on these key parameters. It is not clear why there were no significant differences in other variables in relation to perceived stress. The authors did not offer sufficient information in their discussion, particularly addressing the discrepancy between their study and other reports.
Their data also seemed to suggest that resilience was positively correlated with quality of life while quality of life correlated negatively with age and comorbidity. This data was not discussed or explained in the paper. If employment status significantly impact the stress level why didn't it affect resilience and quality of life?
At last, even stress was shown to be correlated negatively with resilience the evidence for interventions should be sought and provided. Resilience could change how one perceive stress but self-perception of stress could affect resilience too hence it is important to establish cause and effect relationship. There is a clear need for more studies.
Author Response
REV2
It is an very interesting study with the potential to provide some very important information which could lead to positive outcomes for chronic kidney diseases patients.
Dear Reviewer,
We greatly appreciate your comments and are pleased that you find this study to be of interest. We are aware that with your insight and suggestions the manuscript has been significantly improved. Your concerns have been addressed and we hope that you consider this manuscript for publication.
This manuscript has been reviewed by a professional translator in order to improve and facilitate the understanding of the article. We hope that the resulting changes are to the liking of the reviewers.
However the report only identified significant correlations between employment status, resilience, quality of life and perceived stress while many other variables showed no impact on these key parameters. It is not clear why there were no significant differences in other variables in relation to perceived stress. The authors did not offer sufficient information in their discussion, particularly addressing the discrepancy between their study and other reports.
We appreciate this observation, and to facilitate the understanding of these concepts we have introduced the following paragraph into the discussion:
¨The SSR model helps to explain the relationship between perceived stress, health-related quality of life, and resilience. Resilient resources, emotional regulation strategies (which modulate perceived stress), and resilient beliefs (perceived quality of life) aid in the management of stressful situations. Once the stressful event has been overcome, in the short term the use of resources and new emotional regulation strategies can modify perceived quality of life. In the medium term and after a period of self-reflection, resilient resources, their use, and resilient values are redefined, finally modifying resilience [14]. Finally, stable resilient resources (sex, age, living arrangement, educational level, or home setting) then have a reduced influence on stress with each new situation. This could explain the absence of significant relationships between stress and other socio-demographic, HD routine-related, or clinical variables.¨
Crane, M. F., Searle, B. J., Kangas, M., & Nwiran, Y. (2019). How resilience is strengthened by exposure to stressors: The systematic self-reflection model of resilience strengthening. Anxiety, Stress, & Coping, 32(1), 1-17.
Their data also seemed to suggest that resilience was positively correlated with quality of life while quality of life correlated negatively with age and comorbidity. This data was not discussed or explained in the paper. If employment status significantly impact the stress level why didn't it affect resilience and quality of life?
We understand and appreciate this insight with regard to employment status, and given the low participation of actively employed individuals in this study, we cannot reach conclusions on the causality of the relationship of this variable with stress, resilience and quality of life. Thus, we have introduced this sentence into the discussion:
¨Those who remain active in the workplace (with very low representation in our sample) and above all retired individuals have lower levels of perceived stress than the unemployed or those engaged in domestic work, possibly due to better income (typically a protective factor against stress) and because they maintain higher levels of activity and socialization (also protective against stress) [34]. The loss of resources has been found to be related to higher levels of perceived stress [35-38], emphasizing the importance of an intersectoral approach and social interventions to reinforce possible unfavorable situations in these patients [39]. With the results obtained in our sample, it was not possible to explain the fact that while employment status influenced stress, it did not affect resilience and quality of life.¨
At last, even stress was shown to be correlated negatively with resilience the evidence for interventions should be sought and provided. Resilience could change how one perceive stress but self-perception of stress could affect resilience too hence it is important to establish cause and effect relationship. There is a clear need for more studies.
Thank you for this reflection on our work. To facilitate the understanding of these concepts we have introduced the following paragraph into the introduction:
¨Among the models aimed at understanding the fundamental elements of resilience is the Systematic Self-Reflection (SSR) model. This model differentiates three basics concepts for coping: (a) resilient resources, (b) their use and emotional regulation, and (c) resilient beliefs [14]. Resilient resources in HD include the cognitive, employment-related, family, and economic resources used to cope with the disease.¨
This paragraph has also been introduced into the discussion:
The SSR model helps to explain the relationship between perceived stress, health-related quality of life, and resilience. Resilient resources, emotional regulation strategies (which modulate perceived stress), and resilient beliefs (perceived quality of life) aid in the management of stressful situations. Once the stressful event has been overcome, in the short term the use of resources and new emotional regulation strategies can modify perceived quality of life. In the medium term and after a period of self-reflection, resilient resources, their use, and resilient values are redefined, finally modifying resilience [14]. Finally, stable resilient resources (sex, age, living arrangement, educational level, or home setting) then have a reduced influence on stress with each new situation. This could explain the absence of significant relationships between stress and other socio-demographic, HD routine-related, or clinical variables.
Crane, M. F., Searle, B. J., Kangas, M., & Nwiran, Y. (2019). How resilience is strengthened by exposure to stressors: The systematic self-reflection model of resilience strengthening. Anxiety, Stress, & Coping, 32(1), 1-17.
Reviewer 3 Report
The authors examined factors associated with the perceived stress in patients with end-stage renal disease (ESRD) under hemodialysis. Overall, the idea is not without interest, but several methodological issues need to be addressed.
- Introduction:
- Line 43: …lifestyle modifications that include healthy habits, “stay in the health center for HD, and care of vascular access.” I don’t think the phrases in the quotation marks belong to “lifestyle modifications”. Should be changed.
- Line 44: whit? Also, HD is not the main reason for the unwanted physiological effects such as dry skin, itching,…The authors may have to consult a nephrologist for what they want to describe before writing the manuscript.
- Line 48~49: “One way of delaying the evolution of the disease…is the use of tools to reach a consensus in decision-making between…” This sentence is confusing and not reasonable. Consensus reaching is not believed to alter disease course in chronic kidney disease (CKD) patients, but to alter one’s perception of illness, care quality, and others.
- Line 58~70: this paragraph dealt with quality of life and stress origin in CKD patients, but the coherence of this paragraph with the former and the latter one is poor. The HRQoL concept was only lightly put forth and then the concept of stress was brought in. Resilience just came out all of a sudden. I strongly suggest that the authors make their introduction coherent, with a better logic, introduce each concept smoothly with connections, and describe their study hypothesis clearly. The current format does not live up to the standard of an ordinary journal article.
- Line 80: the study aim was laboriously described. Should be described more clearly and not contain all details of the study patients.
- Methods
- Design: “correlational” is really unnecessary.
- Line 91: Why were patients with “ACKD” included? Did this study focus on those with acute kidney injury? This should be explained in detail as the focus and rationale will completely differ if patients with acute kidney injury are enrolled predominantly.
- There were too many patients excluded in this study (144/407). This tend to cause preferential selection of patients for analysis and bias the results. The authors need to provide a Table comparing the enrolled patients and the patients excluded regarding all clinical features to ensure no selection bias (this is mandatory).
- Results:
- Line 145: Only 29% of enrollees had diabetes, a phenomenon that differs substantially with other studies of ESRD patients (DM predominant). Please explain and provide evidence that no selection bias occurs.
- The authors stated that normality tests have been done. Please provide the Kolmogorov-Smirnov and Shapiro-Wilk normality tests p values for all continuous variables in a new table and justify the subsequent analyses accordingly.
- The definition of comorbidity is unclear. Is comorbidity a continuous variable? (This is very unlikely in my view)
- Please provide the model stability measures of the linear regression models provided. Without this the regression model cannot be trusted as being valid.
- The English style of this manuscript is very poor and many sentences are confusing. For example, in abstract, “chronic kidney disease requires various stages of…”. However, the disease per se does not require therapy; it is the patients that carry chronic kidney disease require therapy. Many other places with confusing presentations. Please have the manuscript edit by a professional service to increase readability; otherwise, the content is problematic.
Author Response
REV3
Dear Reviewer,
We greatly appreciate your comments. We are aware that with your insight and suggestions the manuscript has been significantly improved. We have addressed your concerns and hope you consider this manuscript for publication.
Introduction:
Line 43: …lifestyle modifications that include healthy habits, “stay in the health center for HD, and care of vascular access.” I don’t think the phrases in the quotation marks belong to “lifestyle modifications”. Should be changed.
We thank you for pointing this out. We have revised this sentence and have made modifications for improved reading as follows:
¨Therapeutic management of the CKD undergoing HD patient is based on pharmacological measures as well as lifestyle modifications or changes in routine that include healthy habits, visits to the health center for HD, and vascular access care [4].¨
Line 44: whit? Also, HD is not the main reason for the unwanted physiological effects such as dry skin, itching,…The authors may have to consult a nephrologist for what they want to describe before writing the manuscript.
To improve this sentence and avoid confusion about the origin of the symptomatology we have made the following modifications:
¨The patient undergoing HD can cause may experience unwanted physiological effects symptoms such as dry skin, itching, tiredness, or loss of energy, which can affect activities of daily living and self-care [5,6].¨
Line 48~49: “One way of delaying the evolution of the disease…is the use of tools to reach a consensus in decision-making between…” This sentence is confusing and not reasonable. Consensus reaching is not believed to alter disease course in chronic kidney disease (CKD) patients, but to alter one’s perception of illness, care quality, and others.
This paragraph has been omitted to improve the introduction, and, as indicated in the following recommendation, to introduce the concepts more smoothly.
Line 58~70: this paragraph dealt with quality of life and stress origin in CKD patients, but the coherence of this paragraph with the former and the latter one is poor. The HRQoL concept was only lightly put forth and then the concept of stress was brought in. Resilience just came out all of a sudden. I strongly suggest that the authors make their introduction coherent, with a better logic, introduce each concept smoothly with connections, and describe their study hypothesis clearly. The current format does not live up to the standard of an ordinary journal article.
Thank you for this insight. This issue has been addressed as described in the above assessment.
Line 80: the study aim was laboriously described. Should be described more clearly and not contain all details of the study patients.
After evaluating your assessment, we describe the objective as follows: “For the aforesaid reasons, this study aimed to identify variables related to higher levels of stress and poorer adaptation of chronic kidney disease patients after six months of treatment with HD. A secondary objective was to identify how the studied variables could predict perceived stress in these patients.”
Methods
Design: “correlational” is really unnecessary.
We are in agreement with this observation and have changed the manuscript accordingly.
Line 91: Why were patients with “ACKD” included? Did this study focus on those with acute kidney injury? This should be explained in detail as the focus and rationale will completely differ if patients with acute kidney injury are enrolled predominantly.
Patients with acute kidney disease were not included. Participants were recruited among patients with advanced chronic kidney disease. We believe that there is an error in the interpretation of this selection criterion.
Participants are defined in the following sentence. We have included the modification that is identified in a different colour:
¨ The sample population was selected from all advance chronic kidney disease (ACKD) patients receiving hospital-based hemodialysis at the Diaverum clinics.¨
There were too many patients excluded in this study (144/407). This tend to cause preferential selection of patients for analysis and bias the results. The authors need to provide a Table comparing the enrolled patients and the patients excluded regarding all clinical features to ensure no selection bias (this is mandatory).
In the following paragraph we have explained the reasons for the loss of patients in the study:
¨Of the 407 patients treated at the Diaverum clinics, 60 patients were treated in three small clinics that chose not to participate in the study, 54 had received treatment for less than six months, 42 presented with cognitive problems, and 28 had problems with reading and writing in Spanish. As a result, 223 potential participants for the study remained.¨
Not all of the participants from the potential sample completed the scales that were necessary for inclusion in the final study sample, as explained in the following sentence:
¨In total, 223 patients were invited to participate in the study. Of these, 64.6% completed all the assessment scales, resulting in a final study sample of only 144 patients.¨
Results:
Line 145: Only 29% of enrollees had diabetes, a phenomenon that differs substantially with other studies of ESRD patients (DM predominant). Please explain and provide evidence that no selection bias occurs.
We understand and accept this comment. Data on diabetic status were provided by the management staff at the clinics. All patients meeting inclusion criteria were invited to participate, so the only cause of bias is that of self-selection.
The authors stated that normality tests have been done. Please provide the Kolmogorov-Smirnov and Shapiro-Wilk normality tests p values for all continuous variables in a new table and justify the subsequent analyses accordingly.
We appreciate your observation and have included the p-values of the Kolmogorov–Smirnov and Shapiro–Wilk tests in Table 2.
The correlations studied have been reviewed using Spearman's Rho test values for all studies, as shown in Table 2.
The methodology has been modified, introducing the sentence: "Given the non-normal distribution of all variables except age, correlations were determined using Spearman's rho".
In the results, we have introduced the symbol rho (ρ), which replaces the r of Pearson (r).
Normality |
||||||
|
Kolmogorov-Smirnova |
Shapiro-Wilk |
||||
Statistic |
gl |
Sig. |
Statistic |
gl |
Sig. |
|
age |
,079 |
98 |
,145 |
,982 |
98 |
,204 |
comorb |
,125 |
98 |
,001 |
,966 |
98 |
,012 |
totalcon100 |
,123 |
98 |
,001 |
,944 |
98 |
,000 |
total pass |
,077 |
98 |
,180 |
,961 |
98 |
,005 |
kdtotal |
,093 |
98 |
,034 |
,976 |
98 |
,066 |
a. Correction of signification of Lilliefors |
The definition of comorbidity is unclear. Is comorbidity a continuous variable? (This is very unlikely in my view)
The study of comorbidity as a continuous variable and its applicability to mortality has been widely used since the 1990s. The publication of this tool was made in “Charlson ME, Pompei P, Ales KL, MacKenzie CR: A new method of classifying prognostic comorbidity in longitudinal studies: development and validation. J Chronic Dis 1987; 40(5): 373-383”
Its abbreviated version has facilitated its study and allows patients to be classified with regard to the absence of comorbidity (score of 0-1), low comorbidity (2 points), and high comorbidity (3 points or more). Although it may seem less accurate than the full original version, its prognostic utility is similar in the short term, although there are no studies with long-term follow-up (Robles MJ, et al. Rev Esp Geriatr Gerontol 1998; 33 [Suppl 1]: 154) (Farriols C, et al. Rev Esp Geriatr Gerontol 2004; 39 [Suppl 2]: 43).
Please provide the model stability measures of the linear regression models provided. Without this the regression model cannot be trusted as being valid.
The regression models have been described using the usual, standardized measures to present these models, so we consider that they are sufficiently presented.
The English style of this manuscript is very poor and many sentences are confusing. For example, in abstract, “chronic kidney disease requires various stages of…”. However, the disease per se does not require therapy; it is the patients that carry chronic kidney disease require therapy. Many other places with confusing presentations. Please have the manuscript edit by a professional service to increase readability; otherwise, the content is problematic.
This manuscript has been reviewed now by a professional translator in order to improve and facilitate the understanding of the article. We hope that the resulting changes are to the liking of the reviewers.
Round 2
Reviewer 3 Report
The authors have made substantial revisions to their text, for which this reviewer feels appreciative. However, some of the requested results have not been provided, which render further interpretation difficult. Please see the following comments.
- A comparison between included and the excluded patients with regard to their demographic features should be provided.
- The authors should use AUROC test to satisfy the readers with regard to the validity of their regression analyses, with results provided. The authors argued that their process was standard and no validity test was needed, a statement which should be considered flawed. Regression analyses can always be calibrated and optimized through validity tests.
- The English style of this manuscript remains suboptimal. Perhaps the authors need a professional editing service to help improve the readability of this manuscript.
